# Metal-free photocatalytic cross-electrophile coupling enables C1 homologation and alkylation of carboxylic acids with aldehydes

Stefano Bonciolini[1,7], Antonio Pulcinella[1,7], Matteo Leone [1,2], Debora Schiroli[1,3], Adrián Luguera Ruiz [1,4], Andrea Sorato [1], Maryne A. J. Dubois[5], Ranganath Gopalakrishnan [5], Geraldine Masson [2], Nicola Della Ca' [3], Stefano Protti [4], Maurizio Fagnoni [4], Eli Zysman-Colman [6], Magnus Johansson [5] & Timothy Noël [1] ✉

In contemporary drug discovery, enhancing the sp³-hybridized character of molecular structures is paramount, necessitating innovative synthetic methods. Herein, we introduce a deoxygenative cross-electrophile coupling technique that pairs easily accessible carboxylic acid-derived redox-active esters with aldehyde sulfonyl hydrazones, employing Eosin Y as an organophotocatalyst under visible light irradiation. This approach serves as a versatile, metal-free C(sp³)−C(sp³) cross-coupling platform. We demonstrate its synthetic value as a safer, broadly applicable C1 homologation of carboxylic acids, offering an alternative to the traditional Arndt-Eistert reaction. Additionally, our method provides direct access to cyclic and acyclic β-arylethylamines using diverse aldehyde-derived sulfonyl hydrazones. Notably, the methodology proves to be compatible with the late-stage functionalization of peptides on solid-phase, streamlining the modification of intricate peptides without the need for exhaustive *de-novo* synthesis.

In drug discovery, the 3D structure of proteins is crucial for the success of drugs. The increased use of sp³-hybridized carbon atoms (Fsp³) is key, as it correlates with a drug's effectiveness and safety[1]. This trend, known as 'Escape from Flatland'[2], involves increasing Fsp³ in drugs for better alignment with protein structures, enhancing selectivity and efficacy[3]. This strategy improves target interaction and reduces side effects, balancing effective treatment with minimal negative effects.

Historically, classical cross-coupling reactions have been a linchpin in synthetic chemistry, enabling the straightforward construction of C(sp²)−C(sp²) bonds and thereby propelling the production of planar, biaryl structures. This entrenched reliance on cross-coupling has inadvertently sculpted a discernible bias in small molecule drug design, steering the generation of libraries that predominantly feature structurally analogous, two-dimensional compounds[4]. While there have been laudable strides made within the domain of C(sp³)−C(sp³) cross-coupling, contemporary methodologies are oftentimes plagued by several pragmatic limitations[5,6]. They typically necessitate sizable excesses of one coupling partner and frequently hinge upon non-abundant starting materials, such as air- and moisture-sensitive alkyl organometallics, thereby constraining the reaction scope and

[1]Flow Chemistry Group, Van't Hoff Institute for Molecular Sciences (HIMS), University of Amsterdam, Science Park 904, 1098 XH Amsterdam, The Netherlands. [2]Institut de Chimie des Substances Naturelles, CNRS, Univ. Paris-Saclay, 1 Avenue de la Terrasse, 91198 Gif-sur-Yvette, Cedex, France. [3]SynCat Lab, Department of Chemistry, Life Sciences and Environmental Sustainability, University of Parma, 43124 Parma, Italy. [4]PhotoGreen Lab, Department of Chemistry, University of Pavia, 27100 Pavia, Italy. [5]Medicinal Chemistry, Research and Early Development, Cardiovascular, Renal and Metabolism (CVRM), BioPharmaceuticals R&D, AstraZeneca, Gothenburg, Sweden. [6]Organic Semiconductor Centre, EaStCHEM School of Chemistry, Purdie Building, North Haugh University of St Andrews, St Andrews, Fife KY16 9ST, UK. [7]These authors contributed equally: Stefano Bonciolini, Antonio Pulcinella. ✉e-mail: t.noel@uva.nl

practicality in a drug discovery context. Consequently, the quest for alternative strategies that circumvent these limitations while facilitating the construction of three-dimensional molecular structures persists as an imperative in medicinal chemistry research[7].

In recent years, nickel-mediated cross-electrophile (XEC) coupling has emerged as a potent strategy for constructing $C(sp^3)$–$C(sp^3)$ bonds, utilizing various native and bench-stable aliphatic coupling entities, thus circumventing the use of moisture-sensitive organometallic species[8–16]. Despite substantial strides within this sphere, exploiting varied, ubiquitous functional groups such as aldehydes as coupling partners has lingered in a state of underdevelopment. Traditionally, aldehydes have been harnessed as carbonyl electrophiles with Mg or Li-based organometallic species or within Nozaki−Hiyama −Kishi (NHK) type reactivity to yield alcohols[17,18], yet their employment to forge $C(sp^3)$–$C(sp^3)$ bonds via a reductive deoxygenative pathway remains, to our knowledge, uncharted. A pioneering approach, that enables the direct coupling of $sp^2$ and $sp^3$ electrophiles, such as aldehydes and carboxylic acids, heralds an attractive disconnection in the cross-electrophile coupling domain (Fig. 1A). Aryl sulfonyl hydrazones are considered as a bench-stable, activated form of aldehydes due to their known propensity to undergo both radical and polar addition,

ultimately yielding deoxygenated, cross-coupled products upon thermal decomposition of alkylated hydrazide intermediates (Fig. 1B)[19–28]. Utilizing abundant aliphatic carboxylic acids activated as NHPI-based redox-active esters (RAEs) to serve as $sp^3$ electrophiles, and employing visible light-mediated decarboxylation to yield carbon-centered radicals[29–32], we envisioned a trapping mechanism with aldehyde sulfonyl hydrazones to, upon sulfinate and dinitrogen extrusion, afford the coveted product (Fig. 1C). In this study, we realize such a metal-free cross-electrophile coupling, leveraging Eosin Y as an economical organophotocatalyst under visible light irradiation[33].

Illustrating the potential of our synthetic strategy becomes particularly appealing when reflecting upon the strategic C1 homologation of carboxylic acids, traditionally achieved through the Arndt-Eistert reaction[34–37]. Although this protocol, developed in the 1950s, bears chemical reliability, significant limitations persist, particularly those pertaining to the generation, purification, and utilization of toxic and explosive diazomethane, hindering its widespread adoption and applicability. While flow technology has provided a partial answer to these safety challenges[38], a truly general and practical alternative for such transformation has been elusive[39]. Indeed, polar variants such as the Kowalsky Ester homologation suffer from the use of organolithium

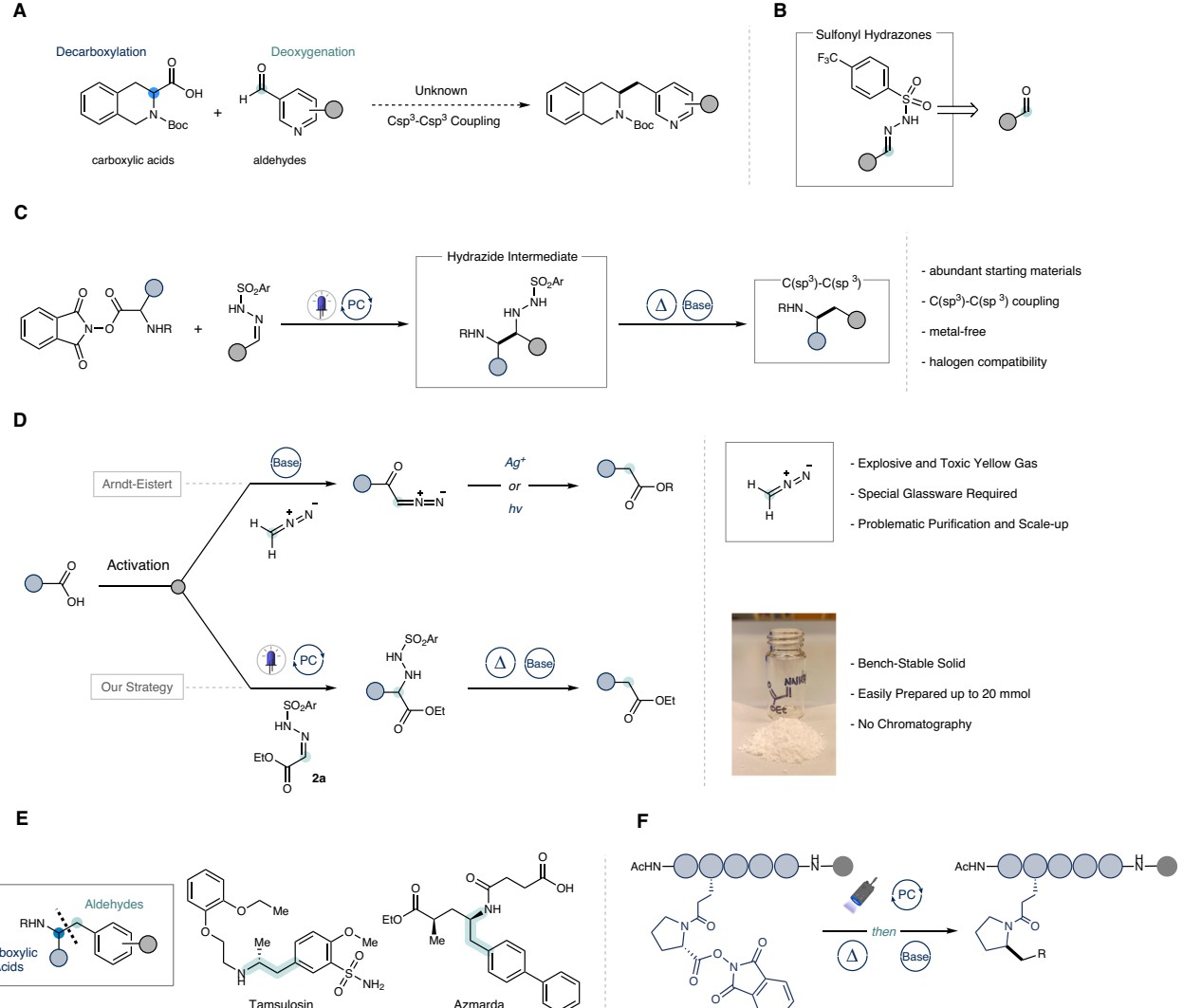

**Fig. 1 | Design and applications of the cross-electrophile coupling of carboxylic acids with aldehydes. A** Elusive cross-electrophile coupling between carboxylic acids and aldehydes. **B** Fast and scalable activation of aldehydes. **C** Our strategy: visible-light promoted coupling of activated carboxylic acids and sulfonyl hydrazones. **D** Application: alternative approach to the classical Arndt-Eistert C1-homologation. **E** Application: retrosynthetic strategy for the synthesis of arylethylamines. **F** Late stage alkylation of peptides on resin.

bases, strongly limiting the substrate scope of the transformation and its scalability[40,41]. For seminal radical variants, Barton proposed a photoinduced C1 homologation of *N*-hydroxy-2-thiopyridone esters, although this strategy suffered from low functional group compatibility, a narrow scope, and requisite lengthy synthetic sequences[42,43]. In this context, we present the utilization of ethyl glyoxalate-derived sulfonyl hydrazone **2a** as a bench-stable and easy-to-handle crystalline radical acceptor to realize the C1 homologation of carboxylic acids under mild conditions (Fig. 1D). As a subsequent, potent application of this synthetic paradigm, our attention was drawn by the synthesis of β-arylethylamines, a prevalent structural motif within numerous drugs and natural products[44]. Although various synthetic routes have been delineated, an intuitive retrosynthetic strategy entailing a cross-coupling reaction between a benzyl electrophile and α-amino nucleophile has remained underrepresented[44–46]. We posit that the advanced cross-electrophile coupling between NHPI esters and aldehyde sulfonyl hydrazones will provide a straightforward and direct route for the efficient preparation of substituted cyclic and acyclic β-arylethylamines (Fig. 1E). Concluding with a third robust synthetic application of this strategy, the methodology demonstrates significant utility in the late-stage functionalization (LSF) of peptides on solid-phase, enabling the modification of complex peptides under mild conditions and obviating the need for tedious de-novo synthesis (Fig. 1F)[47–49].

## Results
### Reaction optimization
We initially commenced to develop a direct decarboxylative C1 homologation, beginning with *N*-Boc (*L*)-Proline, but we were met with failure to produce the desired product **3** (see Supplementary Information, Section 5.1). This result was linked to the noted sensitivity of aldehyde sulfonyl hydrazones to bases, which are indispensable to promote the decarboxylation process[21,50]. Consequently, our investigation focused on the use of well-established *N*-(acyloxy)phthalimides (NHPI-based esters) as redox-active esters (RAEs) in an effort to side-step the necessity for bases during the decarboxylative generation of nucleophilic carbon radicals. An exhaustive screening of all reaction parameters (see Supplementary Information, Section 5.2) led us to discover that the targeted homologated product **3** could be obtained in excellent yields (Table 1, Entry 1, 90% yield) when a dichloromethane (0.1 M) solution composed of ethyl glyoxalate-derived 4-trifluoromethyl-phenyl sulfonyl hydrazone **2a** (1.0 equiv.) as the radical acceptor, *N*-Boc (*L*)-Proline RAE **1a** (1.0 equiv.) as the radical precursor, Hantzsch ester (HE, 1.5 equiv.) as the reductive quencher, and disodium Eosin Y (EYNa₂, 10 mol%) as the photocatalyst was irradiated with blue LEDs (40 W Kessil, 456 nm, PR160L) for 12 h. The yield reflects the one obtained for the final product **3**, achieved when the hydrazinyl intermediate was swiftly subjected to cleavage conditions in ethanol, according to our previous report[27]. Evaluating a two-step one-pot procedure, with trifluorotoluene as the solvent, revealed diminished yields of **3** (Table 1, Entry 2). Surprisingly, an excess of radical acceptor **2a** did not markedly influence the reactivity (Table 1, Entry 3). Noteworthy is the underperformance of more expensive organophotoredox catalysts like 4CzIPN, 3DPA₂FBN or the widely-used transition-metal based photocatalyst Ru(bpy)₃PF₆ (Table 1, Entries 4–6)[51,52]. HE played a major role in the transformation, as other reductive quenchers, such as DABCO, DIPEA, or tetramethylguanidine entirely inhibited the reaction (see Supplementary Table 5).

**Table 1 | Optimization of the photochemical step for the C1 homologation of RAE 1a**

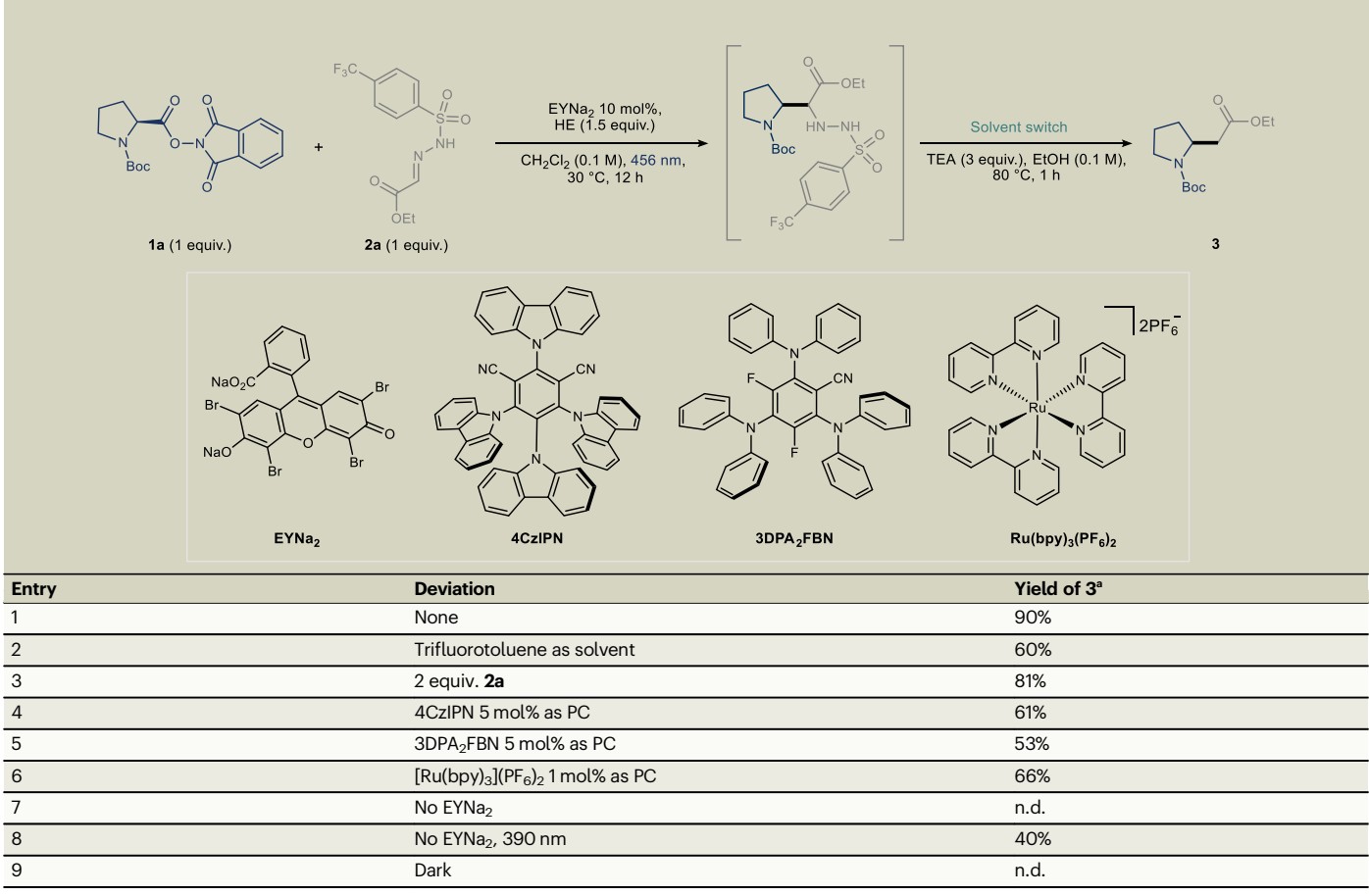

| Entry | Deviation | Yield of 3ᵃ |
|---|---|---|
| 1 | None | 90% |
| 2 | Trifluorotoluene as solvent | 60% |
| 3 | 2 equiv. **2a** | 81% |
| 4 | 4CzIPN 5 mol% as PC | 61% |
| 5 | 3DPA₂FBN 5 mol% as PC | 53% |
| 6 | [Ru(bpy)₃](PF₆)₂ 1 mol% as PC | 66% |
| 7 | No EYNa₂ | n.d. |
| 8 | No EYNa₂, 390 nm | 40% |
| 9 | Dark | n.d. |

ᵃYields were determined by ¹H NMR using trichloroethylene as external standard (0.2 mmol scale, 0.1 M). See Supplementary Information for experimental details.

Fig. 2 | **Scope of the C1 homologation.** Coupling of carboxylic acid-derived redox-active esters (RAEs) with ethyl glyoxylate-derived 4-trifluoromethylphenyl sulfonyl hydrazones **2a**. Reaction conditions: redox active ester (0.3 mmol, 1 equiv.), **2a** (1 equiv.), Hantzsch ester (1.5 equiv.) and EYNa₂ (0.10 equiv.) in 3 mL of CH₂Cl₂ (0.1 M). For further experimental details see the Supplementary Information. [a] > 20:1 d.r. [b] 3:1 d.r. [c] 2.5:1 d.r.

Remarkably, incorporating acidic additives, such as HFIP, TFA, and various amino acids, did not substantially impact the reactivity (see Supplementary Tables 3 and 8). Control experiments conducted to explore the formation of donor-acceptor complexes between RAE **1a** and HE, performed at 456 and 390 nm without EYNa₂, either yielded no product or achieved lower yields (Table 1, Entries 7–8), underscoring the crucial role of the photocatalyst in photoinitiating the reaction, thus securing higher yields[53,54]. Running the reaction in the dark resulted in the quantitative recovery of all starting materials (Table 1, Entry 9). Notably, applying the optimized conditions to the less electrophilic 4-CF₃-benzaldehyde-derived sulfonyl hydrazone **2c** as the radical acceptor yielded the corresponding β-arylethylamine product **46** in a 58% NMR yield. Additional screening of reaction parameters did not produce any enhancements in yield (see Supplementary Information, Section 5.3).

## C1 homologation substrate scope

Having established optimal reaction conditions, we next investigated the scope of the photochemical C1 homologation of RAEs derived from readily available carboxylic acids (Fig. 2). As expected, *N*-Boc protected cyclic amino acids afforded the desired products (**3–5**) in good yields. Moreover, linear proteogenic amino acids underwent homologation to the respective ethyl esters (**6–13**) under the standardized reaction conditions. Noteworthy is the performance of challenging substrates, such as the redox-sensitive methionine and thiophene-derived amino acid, which, despite providing the target

compounds (**8** and **10**), did so in somewhat attenuated yields. The protocol's generality was highlighted through the homologation of sterically hindered cyclic tertiary amino acids, producing the target products in synthetically useful yields (**14–16**). A subsequent examination of various inactivated primary, secondary, and tertiary RAEs revealed that all coupled with glyoxalate-derived sulfonyl hydrazones **2a**, presenting moderate to good yields (**17–22**). In a particularly notable development, two dipeptides underwent photochemical homologation, yielding the targeted homoproline-analogues (**23–24**)[55]. Importantly, the mild conditions of this photocatalytic C1 homologation protocol facilitated the conversion of natural products like biotin and enoxolone—each harboring different sensitive functional groups—to their corresponding ethyl esters (**25–26**), not accessible by the aforementioned methods.

## Alkylation substrate scope

We next aimed to explore further the generality of our developed reaction conditions, applying them to the cross-electrophile coupling of RAEs, derived from a diverse set of carboxylic acids, with various aldehyde-derived sulfonyl hydrazones (Fig. 3). We envisioned providing streamlined access to cyclic and acyclic β-arylethylamines, thereby presenting a new, intuitive radical disconnection for practitioners in the field[44]. Regarding the scope of the α-amino RAEs, a myriad of medicinally pertinent cyclic structures—encompassing azetidine, piperazine, indoline, and isoquinoline—were successfully coupled, achieving synthetically useful yields in all cases (**27–33**)[56].

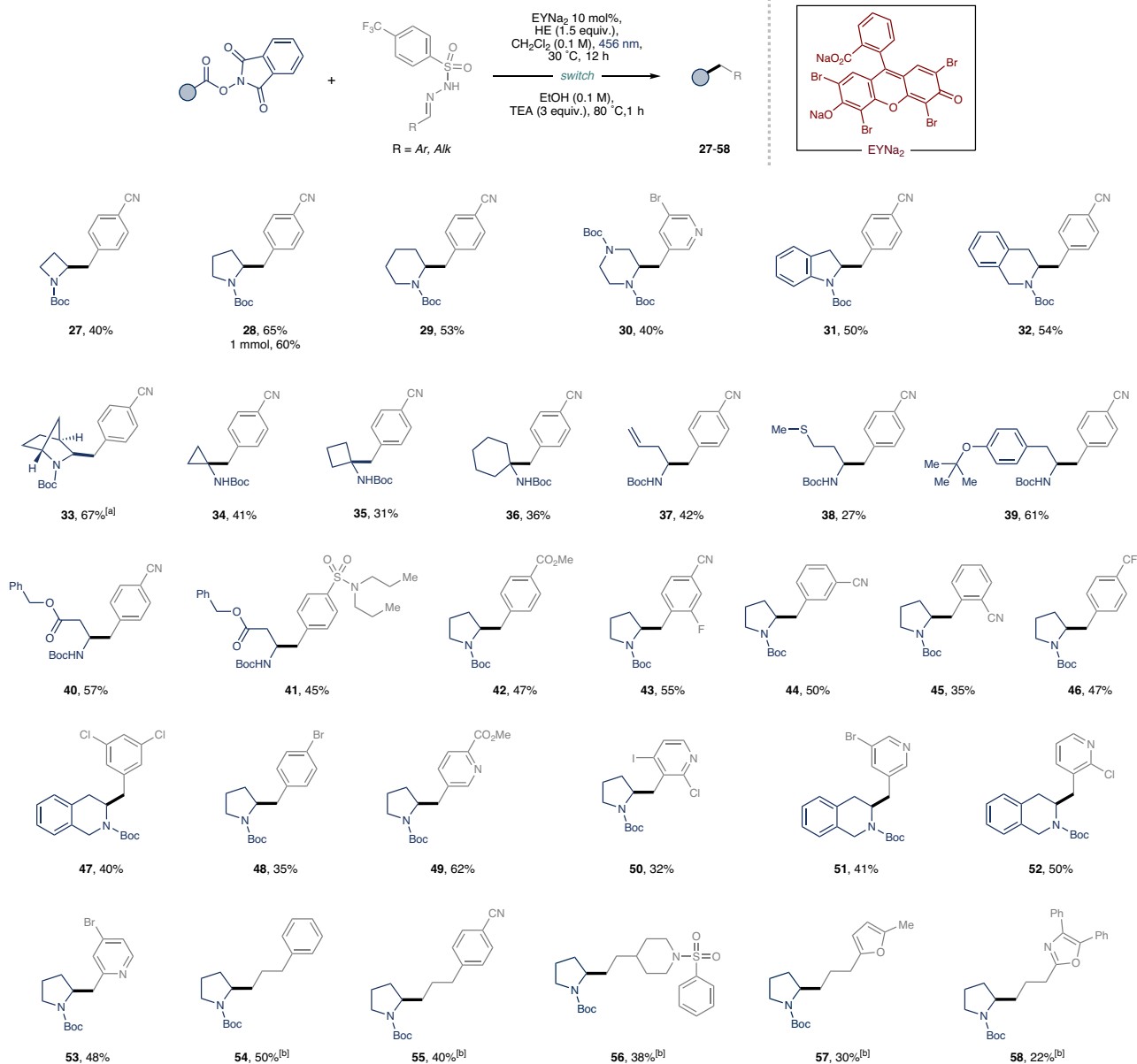

**Fig. 3 | Scope of the alkylation.** Cross-electrophile coupling of RAEs with aromatic and aliphatic aldehyde-derived sulfonyl hydrazones. Reaction conditions: RAE (0.3 mmol, 1 equiv.), Sulfonyl Hydrazone (1 equiv.), Hantzsch ester (1.5 equiv.) and EYNa₂ (0.10 equiv.) in 3 mL of CH₂Cl₂ (0.1 M). For further experimental details see the Supplementary Information. [a] 1:1.4 d.r. [b] 2 equiv. of N-Boc (L)-Proline RAE **1a** was used.

Significantly, the methodology enabled the conversion of even challenging tertiary RAEs, facilitating the creation of quaternary centers, albeit with somewhat reduced yields (**34–36**). Beyond cyclic structures, the protocol also exhibited proficiency with a range of linear amino acids, yielding the corresponding β-arylethylamines in moderate to good isolated yields (**37–41**). An assessment of the sulfonyl hydrazones scope indicated optimal performance with electron-poor groups (see Supplementary Information, Section 11). Noteworthily, the metal-free nature of the protocol tolerated halogenated arenes and heterocycles, providing convenient handles for subsequent synthetic elaboration (**30, 47, 48, 50–53**). A noticeable limitation of the scope was observed: electron-rich sulfonyl hydrazones yielded only traces of the desired product, with a notable reduction of the carboxylic acid. Additionally, under slightly modified reaction conditions (see Supplementary Table 7), unactivated aliphatic aldehyde-derived sulfonyl hydrazones acted as effective coupling partners, delivering alkylated secondary amines in synthetically useful yields, and underlining the method's simplicity and versatility (**54–58**).

## Late-stage modification of peptides on solid phase

Having demonstrated the generality of the photochemical cross-electrophile coupling between sulfonyl hydrazones and RAEs, we turned our inquiry toward the potential extension of this protocol to facilitate the late-stage functionalization (LSF) of more complex molecules, such as peptides. Given the increasing prominence of peptides as therapeutic modalities, the development of methods capable of functionalizing extensive amino acid sequences directly on resin becomes especially valuable, enabling the generation of diversity without necessitating the development of de-novo synthetic methods[57,58]. Moreover, on-resin modification brings forth substantial practical advantages, addressing key challenges related to purification and solubility that are often encountered in peptide chemistry in solution. Specifically, considering the well-documented compatibility

**Fig. 4 | Scope of the cross-electrophile coupling of peptide RAEs on resin.** Reaction conditions: RAE (0.03 mmol, 1 equiv.), Sulfonyl Hydrazone (3 equiv.), Hantzsch ester (4.5 equiv.) and EYNa$_2$ (0.30 equiv.) in CH$_2$Cl$_2$ (33 mM). For full experimental details, see the Supplementary Information.

of redox-active ester synthesis with solid-phase approaches[47,59,60], and the mild basic condition of our two-step protocol, we hypothesized that adapting this photochemical transformation to heterogeneous conditions on resin would be an attainable objective.

At the outset of our investigation, a sensitivity/robustness screening was undertaken to determine which amino acids would be compatible with our reaction conditions and, consequently, could be possibly incorporated into the peptide sequence (see Supplementary Information, Section 5.4). Pleasingly, all screened amino acid residues, when added as additives, did not interfere with the model reaction. Following a minor re-optimization of the reaction parameters and modification of the experimental setup (see Supplementary Information, Sections 7.1–7.4), we discovered that crude peptides, synthesized using Rink Amide resin via SPPS, could be readily engaged in the photocatalytic alkylation (Fig. 4). Illustratively, heptapeptide **P1** was subjected to LSF, yielding the corresponding homoproline-containing analogue **59** in a 28% isolated yield after 21 steps from resin loading (74% LCAP for the decarboxylative alkylation step, with LCAP defined as LC Area % of the product peak in the ultra-performance liquid chromatography (UPLC) chromatogram of the reaction crude. See Supplementary Information, Section 7.5, Supplementary Fig. 20). Highlighting the efficacy of our method, a 28% yield robustly demonstrates the potential of our cross-electrophile coupling for synthesizing complex structures with high selectivity and notable yield conservation. Similarly, a late-stage incorporation of a benzylic unit was accomplished efficiently, demonstrating utility in the context of lipophilicity modulation (**60**) (72% LCAP for the decarboxylative

alkylation step, See Supplementary Information, Section 7.5, Supplementary Fig. 22). To our delight, a derivative of afamelanotide—a therapeutic peptide indicated for patients affected by erythropoietic protoporphyria—was also successfully engaged in the protocol, affording derivative **61** in an overall 9% yield from resin loading (71% LCAP for the decarboxylative alkylation step. See Supplementary Information, Section 7.5, Supplementary Fig. 24)[61].

## Mechanistic investigations

In our pursuit to elucidate the mechanism, we executed a series of experiments to explore the radical pathway and identify the catalytic species facilitating the photochemical transformation. Confirmation of the radical nature of the reaction was achieved through radical trapping and radical clock experiments (Fig. 5A)[62]. Indeed, ESI-HRMS analysis substantiated the formation of TEMPO adduct **62**, while GC-MS analysis convincingly demonstrated carbon radical formation through the production of **64** via a 5-exo-trig radical cyclization.

In light of these observations and based on the reported Single Electron Transfer (SET) mechanism of EYNa$_2$, we propose the ensuing catalytic cycle (see Fig. 5B)[63,64]. Upon absorption of visible light, the triplet excited state of EYNa$_2$ is reductively quenched by the sacrificial electron donor HE to generate HE$^+$. Following the findings of Overmann and König[65–67], the redox-active ester is subsequently reduced by the EYNa$_2$ radical anion, thereby completing the catalytic cycle and yielding the nucleophilic alkyl radical **66** upon decarboxylation. The emergent alkyl radical is then captured by the electrophilic site of sulfonyl hydrazone, resulting in the formation of the hydrazinyl radical

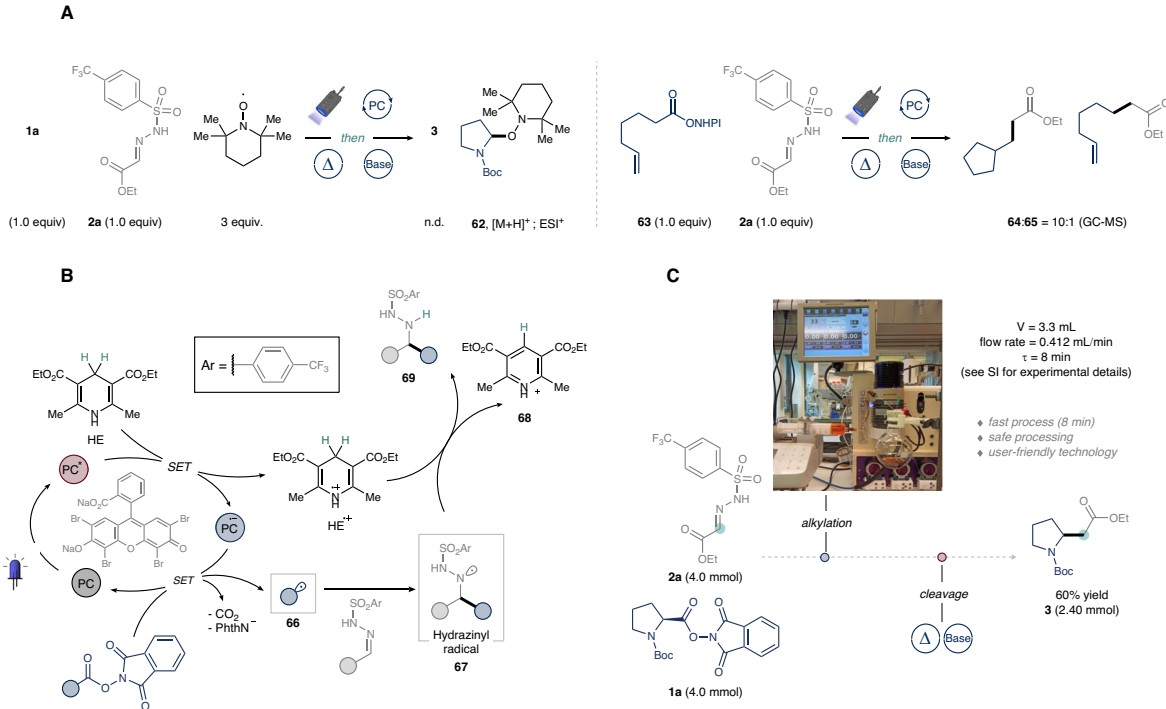

**Fig. 5 | Mechanistic investigation and scale-up in flow. A** Radical trapping and radical clock experiments. **B** Proposed mechanism. **C** Scale-up C1 homologation of **1a** in continuous flow.

intermediate **67**. Finally, a plausible Hydrogen Atom Transfer (HAT) step from HE⁺ or neutral HE to **67** is considered, generating the pyridium co-product **68** and the targeted product **69**.

## Scale up

Finally, we demonstrate the scalability of our photochemical C1 homologation using flow technology (Fig. 5C). In batch settings above 1 mmol, the heterogeneous reaction mixture led to a significant drop in yield of the desired product **3** (see Supplementary Information, Section 9.1). Suspecting non-uniform irradiation and limited light penetration at larger scales, we transitioned the photochemical alkylative step to continuous flow[68–70]. After an extensive optimization conducted at 0.2 mmol scale (see Supplementary Information, Section 9.2), we established conditions for the protocol using a Vapourtec UV-150 photochemical flow reactor (ID: 0.8 mm; $V$ = 3.33 mL, flow rate = 0.412 mL min⁻¹, τ = 8 min) set at 30 °C, irradiated with 60 W 450 nm LEDs. Subsequent thermal cleavage of the alkylated hydrazide intermediate yielded the targeted C1 homologated product in 60% isolated yield.

## Discussion

In summary, we have developed a visible light mediated metal-free cross-electrophile coupling approach that stands as a powerful and versatile C(sp³)–C(sp³) cross-coupling platform. It combines carboxylic acid-derived redox-active esters with aldehyde sulfonyl hydrazones, utilizing Eosin Y as an efficient organophotocatalyst under visible light, leading to the desired cross-coupled products through subsequent fragmentation. Our approach provides a safer alternative to the traditional Arndt-Eistert reaction for C1 homologation of carboxylic acids and enables direct synthesis of cyclic and acyclic β-arylethylamines using diverse aldehyde-derived sulfonyl hydrazones. Furthermore, the method proves also effective for late-stage functionalization (LSF) of peptides on solid-phase. Given these capabilities, we are confident our method will enable the exploration of sp³-hybridized molecules in contemporary drug discovery and development.

## Data availability

The data supporting the results of the article, including optimization studies, experimental procedures, compound characterization, late stage modification of peptides on solid phase, mechanistic studies and scale-up procedures are provided within the paper and its Supplementary Information. Additional data are available from the corresponding author upon request.

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

## Acknowledgements

We are grateful to have received generous funding from the European Union H2020 research and innovation program under the Marie S. Curie Grant Agreement (PhotoReAct, No 956324, S.B., M.L., A.L., G.M., E.Z.C., T.N.; CHAIR, No 860762, A.P., M.J., T.N.). We also would like to thank Ed Zuidinga for help with the HRMS measurements.

## Author contributions

S.B. and A.P. designed the project, with input from T.N. S.B., A.P., M.L., D.S., A.L.R., A.S. performed and analyzed the synthetic experiments with input from G.M., N.D.C. and T.N. The peptide work was carried out by A.P. and was supervised by M.J., M.A.J.D. and R.G. The mechanistic studies were carried out by S.B., and supervised by E.Z.C., S.P., M.F. and T.N. The flow studies were carried out by S.B. and A.P. and supervised by T.N. All authors provided input during the progress meetings. S.B., A.P. and T.N. wrote the manuscript with input from all the authors.

## Competing interests

The authors declare no competing interest.
