## [Peer Review File · Nature Communications]

Metal-free Photocatalytic Cross-Electrophile Coupling enables C1 Homologation and Alkylation of Carboxylic Acids with AldehydesReviewers' Comments:

Reviewer #1:

Remarks to the Author:

The authors report an effective and versatile methodology to generate Csp³-Csp³ bonds via a metal-free, organocatalytic process using visible light irradiation. The process is simple, compatible with a wide range of functionality, and well demonstrated on a variety of highly added-value transformations (C1 homologation, late stage functionalization of complex peptide frameworks). The manuscript is complemented by some mechanistic considerations that further add to an overall remarkable content. It is excellently written and is already at this stage ready for publication.

I would only like to see a concrete justification of every potential author's contribution prior to having them as granted author.

Reviewer #2:

Remarks to the Author:

The present manuscript describes an unprecedented metal free photocatalytic C(sp³)-C(sp³) deoxygenative coupling of carboxylic acids derivatives and sulfonylhydrazones. The authors have developed a mild protocol for the formal C1 homologation of carboxylic acids, significantly expanding the scope of such transformation and overcoming significant limitations associated to the use of hazardous diazomethane (or derivatives). The reaction could be applied to a wide range of carboxylic acids, including natural products and peptides. The synthetic value of the transformation is further highlighted by the application for late stage functionalization of peptides on resin and the possibility of translating the photochemical transformation in flow. Additionally, the authors demonstrated the versatility of such cross electrophile coupling through a new retrosynthetic strategy for the streamlined synthesis of β -arylethylamine. Therefore, I fully recommend its publication on Nature Communications as it meets all the requirements of the journal.

I do have a few minor remarks that the authors should consider:

1. The use of the CF₃ substituent in 4 position on the arylsulfonyl group of the sulfonylhydrazones looks crucial for these transformations. Have you considered using differently substituted sulfonylhydrazones in order to improve reaction efficiency? Specifically, how does the commercially available tosylhydrazine perform in the reaction conditions?
2. How crucial is the role of the solvent in the cleavage step? Have the authors considered other solvents and temperatures and tried to optimize a one pot protocol avoiding the solvent switch?

Reviewer #3:

Remarks to the Author:

This paper reports a novel tactic to couple redox-active esters (RAEs) with aldehyde sulfonyl hydrazones providing an easy access to C1 homologation reactions. The method appears versatile - despite the moderate yields - and compatible with LSF approaches as demonstrated for the functionalization of peptides by solid phase synthesis. The work is well done and represents a significant advance in the field.

I found some issues/questions that need to be addressed before acceptance:

- A general transition-metal-free cross-coupling between benzylic sulfonylhydrazones and alkyl boronic acids was described by Merchant and Lopez in *Org. Lett.* 2020, 22, 6, 2271–2275. The work should be cited and discussed in the text. Also the work by Sakai and MacMillan (*J. Am. Chem. Soc.* 2022, 144, 14, 6185–6192) on the C(sp³)-C(sp³) cross-coupling of alcohols and carboxylic acids by NHC-mediated deoxygenation with hypervalent iodine-mediated decarboxylation, is worth to be cited in the introduction section.

- The solvent seems to play a crucial role for the outcome of the reaction (Table S2 and S3). Authors state '...the pivotal role dichloromethane plays in the photocatalytic cycle' without providing evidences and comments in the 'Mechanistic Investigation' section.

- The reaction scope is good but provides moderate yields in most of the cases (Figure 2-3). Indeed, the higher yield (80%) is obtained with the model substrate 1a. Why the reaction is not equally efficient with other substrates? What about conversion yields? Are there any side/byproducts formed during the reaction?

- I was quite surprised that the reaction was less efficient under flow conditions (80% vs 60%). I suspect the adjustment of the reaction conditions to get flowability significantly affected the reaction efficiency (e.g., use of acetone)?

The main text and references have been revised (and highlighted) in accordance with the formatting guidelines outlined by Nature Communications.

Reviewer 1

The authors report an effective and versatile methodology to generate Csp³-Csp³ bonds via a metal-free, organocatalytic process using visible light irradiation. The process is simple, compatible with a wide range of functionality, and well demonstrated on a variety of highly added-value transformations (C1 homologation, late stage functionalization of complex peptide frameworks). The manuscript is complemented by some mechanistic considerations that further add to an overall remarkable content. It is excellently written and is already at this stage ready for publication.

We kindly thank the Reviewer for these supportive comments.

I would only like to see a concrete justification of every potential author's contribution prior to having them as granted author.

We have already included an author contribution summary in the main article. We understand that the author list is lengthy, reflecting the diverse expertise required to complete this paper. However, we want to assure you that each individual listed as a co-author has made valuable contributions deserving of recognition.

Reviewer 2

The present manuscript describes an unprecedented metal free photocatalytic C(sp³)-C(sp³) deoxygenative coupling of carboxylic acids derivatives and sulfonylhydrazones. The authors have developed a mild protocol for the formal C1 homologation of carboxylic acids, significantly expanding the scope of such transformation and overcoming significant limitations associated to the use of hazardous diazomethane (or derivatives). The reaction could be applied to a wide range of carboxylic acids, including natural products and peptides. The synthetic value of the transformation is further highlighted by the application for late stage functionalization of peptides on resin and the possibility of translating the photochemical transformation in flow. Additionally, the authors demonstrated the versatility of such cross electrophile coupling through a new retrosynthetic strategy for the streamlined synthesis of β -arylethylamine. Therefore, I fully recommend its publication on Nature Communications as it meets all the requirements of the journal. I do have a few minor remarks that the authors should consider:

We kindly thank the Reviewer for these encouraging comments.

1. The use of the CF₃ substituent in 4 position on the arylsulfonyl group of the sulfonylhydrazones looks crucial for these transformations. Have you considered using differently substituted sulfonylhydrazones in order to improve reaction efficiency? Specifically, how does the commercially available tosylhydrazine perform in the reaction conditions?

We thank the Reviewer for this observation. The use of glyoxylate-derived tosyl hydrazone **2i** was attempted but the corresponding hydraziny intermediate **Int-i** proved to be recalcitrant toward the cleavage conditions, making it unsuitable for our two step protocol as a C1-homologating reagent. On the other hand the use of the more reactive 4-NO₂ derivatives **2ii** did not afford the intermediate **Int-ii** either.

Using Tosyl Hydrazone: **No Fragmentation of the Intermediate**

Using 4-NO₂ Sulfonyl Hydrazone: **No Product detection, Neither Intermediate**

2. How crucial is the role of the solvent in the cleavage step? Have the authors considered other solvents and temperatures and tried to optimize a one pot protocol avoiding the solvent switch?

Indeed other options were explored, taking into consideration practical and safety aspects. As reported in the manuscript, trifluorotoluene (Table 1) was considered to develop a two-step one-pot procedure, but reduced yields of **3** were obtained. Dichloromethane was also evaluated for the cleavage step by conducting the reaction in a pressurized tube heated to 80 °C. Unfortunately no product formation was detected under this condition. The controlled fragmentation of the hydraziny intermediate occurs also in the high boiling solvent *n*-butanol, offering subsequent benefits for industrial applications (See Supporting Information, Section 5.2 Optimization of the C1 Homologation Reaction, Table S3). Additionally, when using DMF as solvent for the photochemical step, a one-pot protocol was adopted, avoiding the solvent switch to ethanol (See Supporting Information, Section 5.2, Table S2). However, reactions using dichloromethane and a subsequent solvent switch to ethanol proved generally higher yielding and thus was further used for evaluation of the scope.

Reviewer 3

This paper reports a novel tactic to couple redox-active esters (RAEs) with aldehyde sulfonyl hydrazones providing an easy access to C1 homologation reactions. The method appears versatile - despite the moderate

yields - and compatible with LSF approaches as demonstrated for the functionalization of peptides by solid phase synthesis. The work is well done and represents a significant advance in the field.

We kindly thank the Reviewer for these kind comments.

I found some issues/questions that need to be addressed before acceptance:

A general transition-metal-free cross-coupling between benzylic sulfonylhydrazones and alkyl boronic acids was described by Merchant and Lopez in *Org. Lett.* 2020, 22, 6, 2271–2275. The work should be cited and discussed in the text. Also the work by Sakai and MacMillan (*J. Am. Chem. Soc.* 2022, 144, 14, 6185–6192) on the C(sp³)–C(sp³) cross-coupling of alcohols and carboxylic acids by NHC-mediated deoxygenation with hypervalent iodine-mediated decarboxylation, is worth to be cited in the introduction section.

We agree with the Reviewer and we implemented the suggested references in the main text (see ref. 16 and 28 in the text).

The solvent seems to play a crucial role for the outcome of the reaction (Table S2 and S3). Authors state ‘...the pivotal role dichloromethane plays in the photocatalytic cycle’ without providing evidences and comments in the ‘Mechanistic Investigation’ section.

We agree with this comment. The statement has been revised according to your suggestion.

The reaction scope is good but provides moderate yields in most of the cases (Figure 2-3). Indeed, the higher yield (80%) is obtained with the model substrate 1a. Why the reaction is not equally efficient with other substrates? What about conversion yields? Are there any side/byproducts formed during the reaction?

We kindly thank the Reviewer for raising this point. The reported alkylation procedure performs best when employing cyclic α -amino acid RAEs, e.g. compounds 4 and 5 (Figure 2) were isolated with comparable yields. Evidence of this trend is also found in the scope of the benzylation reaction reported in Figure 3, where the majority of cyclic α -amino acids performed with comparable efficiency. As discussed in the main text, other substrates such as linear α -amino acids and aliphatic carboxylic acids afforded the corresponding product with lower yields, with the major by-product isolated (accounting for almost complete mass-balance) being the corresponding reduced alkane (decarboxylative reduction). In all cases full conversion of the starting material (RAEs) was observed.

I was quite surprised that the reaction was less efficient under flow conditions (80% vs 60%). I suspect the adjustment of the reaction conditions to get flowability significantly affected the reaction efficiency (e.g., use of acetone)?

We agree with the Reviewer assessment. Indeed, the solvent used under flow conditions performed with less efficiency. However, this was also the case in batch (65% vs 80%). Switching the solvent system proved essential to guarantee flowability (i.e., complete solubility of the reaction mixture to avoid clogging). Additionally, as reported in the Supporting Information (See Section 9.2 Scale-up in Continuous Flow) other mixture of solvents completely shut down the desired reactivity. Despite the reduced efficiency, the flow protocol performs generally better than the batch variant when performed at ≥ 1 mmol scale (See Section 9.1) and ensures scalability of the chemistry.

Reviewers' Comments:

Reviewer #2:

Remarks to the Author:

The revisions have addressed all my concerns. Thus, I recommend the publication of this work in Nature Communications.

Reviewer #3:

Remarks to the Author:

Authors have responded to all the issues raised by Referees, therefore I recommend publication of the article in Nature Communications in the current form.